# Projection in genomic analysis: A theoretical basis to rationalize tensor decomposition and principal component analysis as feature selection tools

**Y-h. Taguchi**[1]*, **Turki Turki**[2]

**1** Department of Physics, Chuo University, Bunkyo-ku, Tokyo, Japan, **2** Department of Computer Science, King Abdulaziz University, Jeddah, Saudi Arabia

* tag@granular.com

**Data Availability Statement:** All the data sets and source code are available in GitHub repositry https://github.com/tagtag/peoj.

**Funding:** Japan Society for the Promotion of Science http://dx.doi.org/10.13039/501100001691

## Abstract

Identifying differentially expressed genes is difficult because of the small number of available samples compared with the large number of genes. Conventional gene selection methods employing statistical tests have the critical problem of heavy dependence of $P$-values on sample size. Although the recently proposed principal component analysis (PCA) and tensor decomposition (TD)-based unsupervised feature extraction (FE) has often outperformed these statistical test-based methods, the reason why they worked so well is unclear. In this study, we aim to understand this reason in the context of projection pursuit (PP) that was proposed a long time ago to solve the problem of dimensions; we can relate the space spanned by singular value vectors with that spanned by the optimal cluster centroids obtained from K-means. Thus, the success of PCA- and TD-based unsupervised FE can be understood by this equivalence. In addition to this, empirical threshold adjusted $P$-values of 0.01 assuming the null hypothesis that singular value vectors attributed to genes obey the Gaussian distribution empirically corresponds to threshold-adjusted $P$-values of 0.1 when the null distribution is generated by gene order shuffling. For this purpose, we newly applied PP to the three data sets to which PCA and TD based unsupervised FE were previously applied; these data sets treated two topics, biomarker identification for kidney cancers (the first two) and the drug discovery for COVID-19 (the thrid one). Then we found the coincidence between PP and PCA or TD based unsupervised FE is pretty well. Shuffling procedures described above are also successfully applied to these three data sets. These findings thus rationalize the success of PCA- and TD-based unsupervised FE for the first time.

## Introduction

In genomic sciences, selecting a limited number of differentially expressed genes (DEGs) among as many as several tens of thousands of genes is a critical problem. Unfortunately, this

KAKENHI [grant numbers 19H05270, 20H04848, and 20K12067] Professor Y-h. Taguchi The funders had no role in study design, data collection and analysis, decision to publish, or preparation of the manuscript.

**Competing interests:** The authors have declared that no competing interests exist.

is a very difficult task as the number of genes, $N$, is usually much larger than the number of available samples, $M$. However, as this is not a mathematically solved problem, it has most frequently been tackled empirically using statistical test-based feature selection strategies [1, 2]. Despite huge efforts along this direction, these statistical test-based feature selection strategies cannot be said to work well.

Selection of biologically informative genes including DEGs is essentially performed as follows (For simplicity, $\sum_i x_{ij} = 0$, $\sum_i x_{ij}^2 = N$ and $M$ samples are composed of multiple classes having an equal number of samples). Suppose that we have properties $\boldsymbol{y} \in \mathbb{R}^M$ attributed to $M$ samples. We would like to relate a matrix form of some omics data, e.g., gene expression profiles, $X \in \mathbb{R}^{N \times M}$ to $\boldsymbol{y}$. The overall purpose is to derive $\boldsymbol{b} \in \mathbb{R}^N$ whose absolute values represent the importance of the $i$th gene. The first and the most popular strategy outside genomic sciences is a regression strategy that requires minimization of

$$(\boldsymbol{y} - \boldsymbol{b}X)^2 \tag{1}$$

resulting in

$$\boldsymbol{b} = \boldsymbol{y}X^T(XX^T)^{-1}. \tag{2}$$

The regression approach, Eq (2), is less popular in genomic sciences than in other scientific fields, possibly because of $N \gg M$, which always results in exactly $(\boldsymbol{y} - \boldsymbol{b}X)^2 = 0$ with an infinitely large number of $\boldsymbol{b}$. Thus, it is useless to select a limited number of important features among the total $N$ features. Although adding the regulation term of $L_2$ norm to Eq (1) as

$$(\boldsymbol{y} - \boldsymbol{b}X)^2 + \lambda \boldsymbol{b}^2 \tag{3}$$

with the positive constant $\lambda > 0$ enables selection of a unique $\boldsymbol{b}$ by minimizing Eq (3) as

$$\boldsymbol{b} = \boldsymbol{y}X^T(XX^T + \lambda I)^{-1}, \tag{4}$$

because it does not satisfy $(\boldsymbol{y} - \boldsymbol{b}X)^2 = 0$ anymore, it is not an ideal solution. Although the solution using the Moore-Penrose Pseudoinverse [3]

$$\boldsymbol{b} = \boldsymbol{y}X^\dagger \tag{5}$$

might be better as it satisfies $(\boldsymbol{y} - \boldsymbol{b}X)^2 = 0$ under the condition of $\min_{\boldsymbol{b}} \boldsymbol{b}^2$, it is unclear whether $\min_{\boldsymbol{b}} \boldsymbol{b}^2$ is a good constraint from the biological viewpoint. Adding the regulation term of $L_1$ norm [4] to Eq (1)

$$(\boldsymbol{y} - \boldsymbol{b}X)^2 + \lambda|\boldsymbol{b}| \tag{6}$$

can yield at most $M$ variables, which is not effective when $N \gg M$, because variables larger than $M$ might be biologically informative and should not be neglected. Moreover, addition of $L_1$ norm is known to be a poor strategy when $X$ is not composed of independent vectors, which are very common in genomic science.

The second strategy is a projection strategy

$$\boldsymbol{b} = \boldsymbol{y}X^T \tag{7}$$

that is equivalent to the maximization of

$$\boldsymbol{y} \cdot \boldsymbol{b}X - \frac{1}{2}\boldsymbol{b}^2 \tag{8}$$

and is employed in PCA- and TD-based unsupervised FE (see below). Through the concept of

projection pursuit [5] (PP), it is understood that seeking the projection vector $\boldsymbol{b}$ maximizes interestingness

$$H(\boldsymbol{b}X) \tag{9}$$

which is Eq (8) in this study. As $H(\boldsymbol{b}X)$ is a function of $\boldsymbol{b}$, it is also denoted as $I(\boldsymbol{b})$, which is called projection index. $I(\boldsymbol{b})$ can be any other function, but its selection should be decided such that the biologically most meaningful results are obtained. Upon obtaining $\boldsymbol{b}$ that maximizes $I(\boldsymbol{b})$, we can select $i$ having a larger absolute $b_i$ as mentioned above. In the framework of PP, in a high dimensional system, almost all $\boldsymbol{b}$ have finite projections [6]. Thus, the only the point is if it is accidental or biologically meaningful.

In genomic science, projection strategy, Eq (7), is also unpopular. Although the reason for the unpopularity of the projection strategy, Eq (7), is unclear, this may be explained by the ignorance of the contribution perpendicular to $\boldsymbol{y}$, $|(\boldsymbol{x}_i \cdot \hat{\boldsymbol{y}})\hat{\boldsymbol{y}} - \boldsymbol{x}_i|$, where $\hat{\boldsymbol{y}}$ is a unit vector parallel to $\boldsymbol{y}$ and is defined as $\boldsymbol{y}/|\boldsymbol{y}|$. Nevertheless, in contrast to the regression strategy requiring the computation of $(XX^T)^{-1}$, Eq (7) can be always computable even if $N \gg M$, which is a great advantage of the projection strategy when compared with the regression strategy.

Instead of these two strategies, feature selection based on statistical tests [1, 2] is more popular in genomic sciences as mentioned above. They try to identify genes whose expression is significantly distinct between classes. Despite its popularity, feature selection based on statistical tests has critical problems; in particular, significance is heavily dependent on sample size, $M$. Even in the case of a small distinction, more significant results are obtained when more samples are considered; this is not applicable biologically because determination of whether gene expression between classes differs significantly should not be a function of sample size. To compensate this heavy sample dependence of significance, other criteria such as fold change between classes are often employed. Thus, feature selection based on statistical tests is at best, the best among the worst approaches. If better strategies can be employed, there will be no reason to employ strategies based on statistical tests.

Despite the unpopularity of projection strategy, it was sometimes evaluated as more effective [7, 8] than the standard feature selection strategy based on statistical tests. Thus, it can be a candidate strategy that can be replaced with feature selection based on statistical tests. In this paper, we try to understand why PCA-based unsupervised FE and TD-based unsupervised FE [3] are effective in feature selection based on projection strategy, since PCA-like as well as TD-like methods were successfully applied in other fields, too [9–11]. We consider the cases biomarker identification of kidney cancer [12] as well as SARS-CoV-2 infection problem [13]; in these studies, despite unsuccessful results obtained by conventional feature selection based on statistical tests, TD-based unsupervised FE identified biologically reasonable genes (for more details about how PCA- and TD-based unsupervised FE are superior to statistical test-based feature selection tools in these specific examples, see these previous studies [12, 13]).

## Materials and methods

Sample R cods is available in https://github.com/tagtag/peoj.

## Expression profiles

mRNA, miRNA, and gene expression profiles in the first, second, and third data sets can be downloaded from TCGA as well as GEO. Their availability is described in detail in previous studies [12, 13].

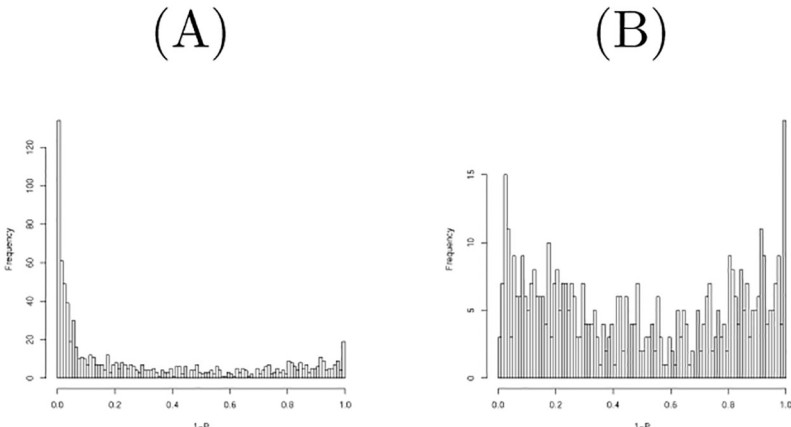

**Fig 1. Histogram of raw *P*-values computed using the null distribution generated by shuffling when miRNAs in the first data set were considered.** (A) All miRNAs (B) Top 500 most expressive miRNAs.

## Excluding low expressed miRNAs, mRNAs, and genes

To draw Figs 1(B), 2(B) and 3(B), low expressed miRNAs, mRNAs, and genes were screened out. For this, we rank them using $\Sigma_j |x_{ij}|$, $\Sigma_j |x_{ik}|$, $\Sigma_{jkm} |x_{ijkm}|$ and only selected the top ranked ones.

## QQplot

QQplot [14] was used to visualize the coincidence between two distributions that do not always have same number of elements. The `qqplot` function implemented in R [15] was employed to draw QQplots (Figs 4 and 5) in this study.

## Null distribution

The null distributions used for computing *P*-values in Figs 1–3 and 6 were generated by gene order shuffling as follows. First, the order of *i* was shuffled within each $x_{ij}$ or within each $x_{ijkm}$ and that of *k* was shuffled within each $x_{kj}$. Thus, the order of mRNAs, miRNAs, and genes was

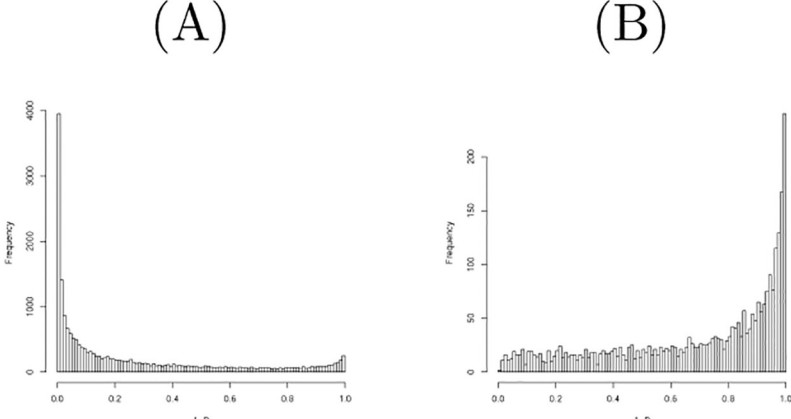

**Fig 2. Histogram of raw *P*-values computed using the null distribution generated by shuffling when the mRNAs in the first data set were considered.** (A) All mRNAs (B) Top 3000 most expressive mRNAs.

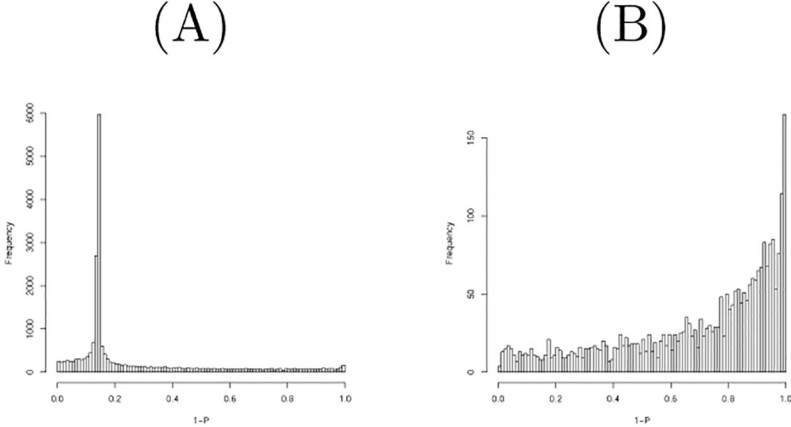

**Fig 3. Histogram of raw *P*-values computed using the null distribution generated by shuffling when genes in the third data set were considered.** (A) All genes (B) Top 2780 most expressive genes.

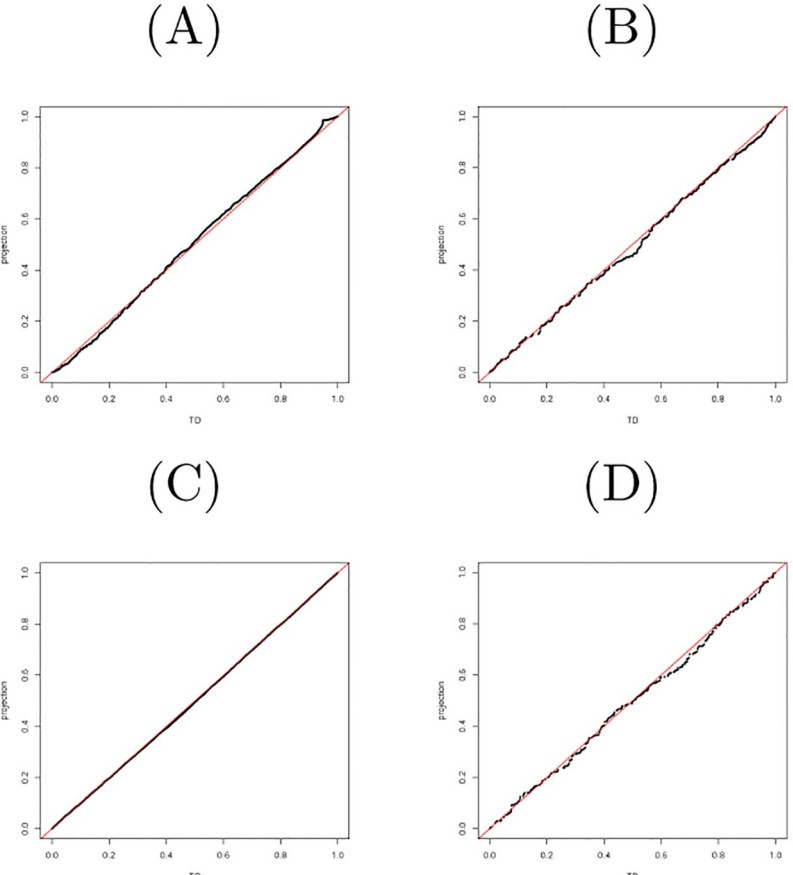

**Fig 4. QQplot between *P*-values computed by TD-based unsupervised FE and projection (A) mRNA in the first data set (B) miRNA in the first data set (C) mRNA in the second data set (D) miRNA in the second data set.**

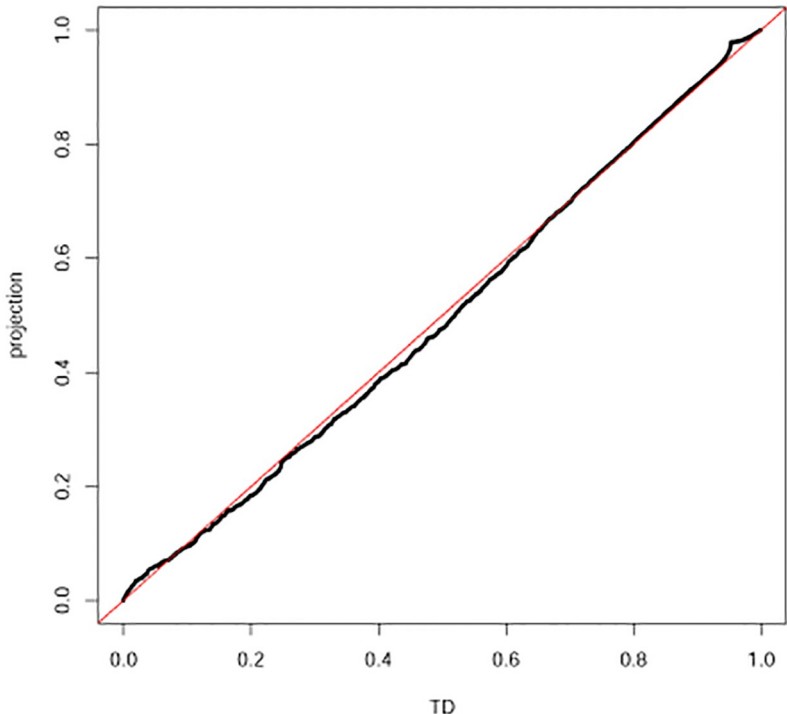

**Fig 5. QQplot of *P*-values between TD-based unsupervised FE and PP (the third data set).**

shuffled such that they differed between samples. Then SVD or TD was applied to $x_{ijk}$ or $x_{ijkm}$ and $u_{2i}$ and $u_{2k}$ from SVD and $u_{5i}$ from TD were generated one hundred times. The null distributions were composed of the generated singular value vectors and *P*-values were computed.

## Results

Fig 7 shows the work flow of this study. In PP, the projection direction is predefined by *y* in a supervised manner while if we do not want to set projection directions in advance we can use

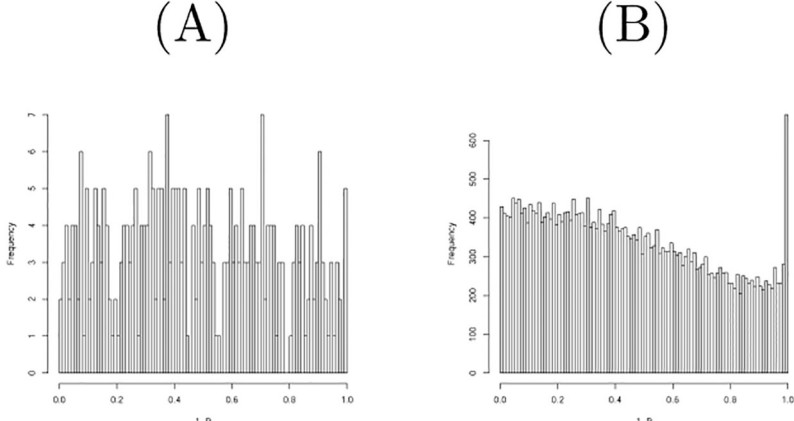

**Fig 6. Histogram of raw *P*-values computed using the null distribution generated by shuffling when the second data set were considered.** (A) All miRNAs (B) All mRNAs.

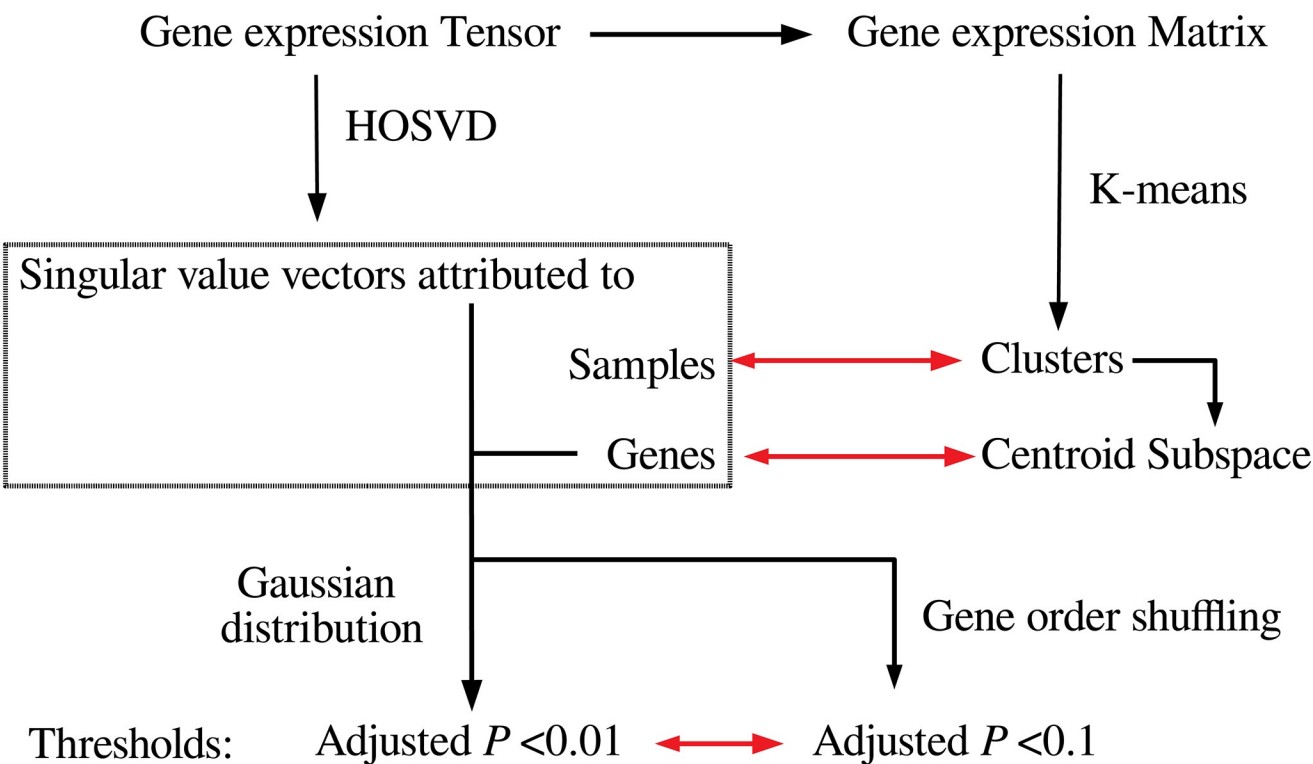

**Fig 7. Discussion of work flow used in this study.** Tensor decomposition (HOSVD) was applied to tenors and using obtained singular value vectors assumed to obey Gaussian distribution, *P*-values are attributed to genes. The genes associated with adjusted *P*-values less than 0.01 are selected. *P*-values are also computed by shuffling and the genes associated with adjusted *P*-values less than 0.1 are well coincident with the genes selected by HOSVD. The correspondence between singular value vectors and K-means applied to unfolded matrices is also discussed.

those determined by PCA or TD, which we call unsupervised FE. There are some advantages of PCA and TD, which are not shared with PP. For example, projection directions not related to the label $\boldsymbol{y}$ may have additional information. In that case, PCA and TD can capture what PP cannot. PCA and TD can be applicable even if pre-defined $\boldsymbol{y}$ is not provided. Thus, PCA and TD have more potential to be applied to wide range of data sets that PP.

**PCA-based unsupervised FE**

Before starting to rationalize PCA- and TD-based unsupervised FE, we briefly summarize how they work. The purpose of PCA- and TD-based unsupervised FE is to select biologically sound features (typically genes) based on the given omics data such as gene expression profiles, in an unsupervised manner. In this subsection, we introduce PCA-based unsupervised FE; TD-based unsupervised FE is an advanced version of PCA-based unsupervised FE and will be introduced in the next subsection.

Suppose that we have gene expression data in a matrix form, $X \in \mathbb{R}^{N \times M}$ for $N$ genes measured across $M$ samples. First, we need to standardize $X$ as $\sum_i x_{ij} = 0$ and $\sum_i x_{ij}^2 = N$ as we will attribute principal component (PC) scores to genes whereas PC loading will be attributed to samples. The $\ell$th PC score attributed to the $i$th gene, $u_{\ell i}$, can be obtained as the $i$th component of the $\ell$th eigenvector, $\boldsymbol{u}_\ell \in \mathbb{R}^N$, of a gram matrix $XX^T \in \mathbb{R}^{N \times N}$, where $X^T$ is a transpose matrix of $X$, as

$$XX^T \boldsymbol{u}_\ell = \lambda_\ell \boldsymbol{u}_\ell \tag{10}$$

where $\lambda_\ell$ is the $\ell$th eigenvalue. Further, the $\ell$th PC score attributed to the $j$th sample, $v_{\ell j}$, can be obtained as the $j$th component of the vector $\boldsymbol{v}_\ell \in \mathbb{R}^M$ defined as

$$\boldsymbol{v}_\ell = X^T \boldsymbol{u}_\ell. \tag{11}$$

Notably, $\boldsymbol{v}_\ell$ is also an eigenvector of the covariance matrix, $X^T X \in \mathbb{R}^{M \times M}$ because

$$X^T X \boldsymbol{v}_\ell = X^T X X^T \boldsymbol{u}_\ell = X^t \lambda_\ell \boldsymbol{u}_\ell = \lambda_\ell \boldsymbol{v}_\ell. \tag{12}$$

PCA-based unsupervised FE works as follows. First, we need to identify the $\boldsymbol{v}_\ell$ of interest. The $\boldsymbol{v}_\ell$ of interest depends on the problem. It might be the one coincident with the samples cluster, or the one with monotonic dependence on some external parameter such as time. After identifying the $\boldsymbol{v}_\ell$ of interest, we try to attribute $P$-values to genes assuming that the components of the corresponding $\boldsymbol{u}_\ell$ follow a normal distribution

$$P_i = P_{\chi^2}\left[ > \left(\frac{u_{\ell i}}{\sigma_\ell}\right)^2 \right] \tag{13}$$

where $P_{\chi^2}[> x]$ is the cumulative $\chi^2$ distribution that the argument is larger than $x$ and $\sigma_\ell$ is the standard deviation. Computed $P$-values are adjusted based on the BH criterion [3] and features associated with adjusted $P$-values less than a specified threshold value can be selected. The reason for the proper working of such a simple procedure is explained later.

Finally, we would like to emphasize the equivalence between singular value decomposition (SVD) and PCA. Suppose we have the SVD of $X$ as

$$x_{ij} = \sum_{\ell=1}^{\min(N,M)} \lambda_\ell u_{\ell i} v_{\ell j}. \tag{14}$$

It is straight forward to show

$$XX^T \boldsymbol{u}_\ell = \lambda_\ell \boldsymbol{u}_\ell \tag{15}$$

$$X^T X \boldsymbol{v}_\ell = \lambda_\ell \boldsymbol{v}_\ell \tag{16}$$

where $\boldsymbol{u}_\ell = (u_{\ell 1}, u_{\ell 2}, \cdots, u_{\ell N})^T$ and $\boldsymbol{v}_\ell = (v_{\ell 1}, v_{\ell 2}, \cdots, v_{\ell M})^T$. Thus, SVD and PCA are mathematically equivalent problems.

## TD-based unsupervised FE

TD-based unsupervised FE works quite similar to PCA-based unsupervised FE. Instead of PCA, we apply TD to $x_{ijk} \in \mathbb{R}^{N \times M \times K}$, that is, for example, the expression of the $i$th gene measured in the $k$th tissue of the $m$th person (even though we consider a three-mode tensor here, extension to the higher mode tensor is straightforward). To obtain TD, we specify the higher-order singular decomposition [3] (HOSVD) as

$$x_{ijk} = \sum_{\ell_1}\sum_{\ell_2}\sum_{\ell_3} G(\ell_1\ell_2\ell_3) u_{\ell_1 j} u_{\ell_2 k} u_{\ell_3 i} \tag{17}$$

where $G(\ell_1\ell_2\ell_3) \in \mathbb{R}^{M \times K \times N}$ is a core tensor, and $u_{\ell_1 j} \in \mathbb{R}^{M \times M}$, $u_{\ell_2 k} \in \mathbb{R}^{K \times K}$, $u_{\ell_3 i} \in \mathbb{R}^{N \times N}$ are singular value matrices. After identifying the $u_{\ell_1 j}$ and $u_{\ell_2 k}$ of interest, for instance, the distinction between healthy controls and patients as well as tissue specific expression, we seek $\ell_3$ associated with $G(\ell_1\ell_2\ell_3)$ having the largest absolute value given as $\ell_1$, $\ell_2$. Then using the identified $\ell_3$, we

attribute *P*-values to the *i*th feature as in the case of PCA-based unsupervised FE,

$$P_i = P_{\chi^2}\left[ > \left(\frac{u_{\ell_3 i}}{\sigma_{\ell_3}}\right)^2 \right] \tag{18}$$

where $\sigma_{\ell_3}$ is the standard deviation. Computed *P*-values are adjusted based on the BH criterion and features associated with adjusted *P*-values less than a specified threshold value can be selected. The reason for the proper working of such a simple procedure is explained later.

## Rationalization of PCA- and TD-based unsupervised FE

To explain why PCA- and TD-based unsupervised FE work rather well, we consider two recent works [12, 13], in which the superiority of PCA- and/or TD-based unsupervised FE over conventional statistical methods was shown; in these studies, conventional statistical test-based methods failed to select a reasonable number of genes whereas TD-based unsupervised FE successfully selected a biologically reasonable restricted number of genes.

In the first study [12], two independent sets of data including the mRNA and miRNA expression of kidney cancer and normal kidney were analyzed in an integrated manner using PCA as well as TD-based unsupervised FE.

**The first data set.** The first data set comprised $M = 324$ samples including 253 kidney tumors and 71 normal kidney tissues. The expression of $N$ mRNAs and $K$ miRNAs was formatted as matrices as $x_{ij} \in \mathbb{R}^{N \times M}$ and $x_{kj} \in \mathbb{R}^{K \times M}$, respectively. The three mode-tensor $x_{ijk} \in \mathbb{R}^{N \times M \times K}$ was generated as

$$x_{ijk} = x_{ij}x_{kj}. \tag{19}$$

As the data were too large to be loaded into the memory available in a standard stand-alone server, it was impossible to obtain TD

$$x_{ijk} = \sum_{\ell_1}\sum_{\ell_2}\sum_{\ell_3} G(\ell_1 \ell_2 \ell_3) u_{\ell_1 i} u_{\ell_2 j} u_{\ell_3 k}. \tag{20}$$

Instead, we generated

$$x_{ik} = \sum_j x_{ijk} \tag{21}$$

and SVD was applied to $x_{ik}$ as

$$x_{ik} = \sum_{\ell = \ell_1 = \ell_3}^{\min(N,K)} \lambda_\ell u_{\ell_1 i} u_{\ell_3 k} \tag{22}$$

to obtain $u_{\ell_1 i}$ and $u_{\ell_3 k}$ approximately. Missing singular value vectors attributed to mRNA and miRNA samples were approximately recovered using the equations

$$u_{\ell_1 j}^{\mathrm{mRNA}} = \sum_{i=1}^{N} x_{ij} u_{\ell_1 i} \tag{23}$$

$$u_{\ell_3 j}^{\mathrm{miRNA}} = \sum_{k=1}^{K} x_{kj} u_{\ell_3 k} \tag{24}$$

respectively. Although we do not intend to insist that these approximations are precise enough, we decided to employ them as since they turned out to work well empirically. After

investigating the obtained $u_{\ell_{1j}}^{\text{mRNA}}$ and $u_{\ell_{3j}}^{\text{miRNA}}$, we realized that $\ell_1 = \ell_3 = 2$ are coincident with the distinction between tumors and normal tissues; therefore, we attributed $P$-values to mRNA and miRNA using $u_{2i}$ and $u_{2k}$, respectively with the equations

$$P_i = P_{\chi^2}\left[> \left(\frac{u_{2i}}{\sigma_2}\right)^2\right] \tag{25}$$

$$P_k = P_{\chi^2}\left[> \left(\frac{u_{2k}}{\sigma_2'}\right)^2\right]. \tag{26}$$

These $P$-values were corrected by the BH criterion and we selected 72 mRNAs and 11 miRNAs associated with adjusted $P$-values less than 0.01, respectively.

**The second data set.** The second data set comprised $M = 34$ samples including 17 kidney tumors and 17 normal kidney tissues. The same procedures applied to the first data set were also applied to the second data set and we selected 209 mRNAs and 3 miRNAs associated with adjusted $P$-values less than 0.01, respectively. Although various biological evaluations were performed for mRNAs and miRNAs selected using the first data set, the most remarkable achievement was that all three miRNAs selected using the second data set were included in the 11 miRNAs selected using the first data set, and there were as many as 11 common mRNAs selected between the first and second data sets. If we consider that there are as many as several hundred miRNAs and a few tens of thousand mRNAs available, these overlaps are a great achievement as these two data sets are completely independent of each other.

**Comparisons with PP.** To understand why such simple procedures can work well in the framework of PP, we replaced the singular value vectors attributed to samples with projections. For this, we applied PP as mentioned above.

$$y_j = \begin{cases} -\dfrac{M}{M_N}, & j \leq N_N \\[2mm] \dfrac{M}{M_T}, & j > N_N \end{cases} \tag{27}$$

where $M_N$, $M_T$ are the numbers of normal tissues and cancer samples, respectively, and $M_N + M_T = M$. Then we applied PP as

$$b_i = \sum_{j=1}^{M} x_{ij} y_j \tag{28}$$

$$b_k = \sum_{j=1}^{M} x_{kj} y_j. \tag{29}$$

Since $b_i$s and $b_k$s are expected to play the roles of $u_{2i}$ and $u_{2k}$ in Eqs (25) and (26), respectively, we used the absolute values of $b_i$ and $b_k$ to select mRNAs and miRNAs that are presumably coincident with the distinction between tumors and normal tissues. $P$-values are attributed to

**Table 1. Confusion matrix of selected mRNAs between TD-based unsupervised FE and PP in the first data set.** *P*-value computed by Fisher's exact test is $1.90 \times 10^{-149}$.

| | | PP | |
|---|---|---|---|
| | | adjusted $P_i > 0.01$ | adjusted $P_i < 0.01$ |
| TD based unsupervised FE | adjusted $P_i > 0.01$ | 19447 | 17 |
| | adjusted $P_i < 0.01$ | 11 | 61 |

**Table 2. Confusion matrix of selected miRNAs between TD-based unsupervised FE and PP in the first data set.** *P*-value computed by Fisher's exact test is $2.76 \times 10^{-23}$.

| | | PP | |
|---|---|---|---|
| | | adjusted $P_k > 0.01$ | adjusted $P_k < 0.01$ |
| TD based unsupervised FE | adjusted $P_k > 0.01$ | 812 | 2 |
| | adjusted $P_k < 0.01$ | 0 | 11 |

mRNA and miRNA as

$$P_i = P_{\chi^2}\left[ > \left(\frac{b_i}{\sigma_b}\right)^2 \right] \tag{30}$$

$$P_k = P_{\chi^2}\left[ > \left(\frac{b_k}{\sigma_b}\right)^2 \right]. \tag{31}$$

These *P*-values are corrected by the BH criterion and we selected 78 mRNAs and 13 miRNAs for the first data set and 194 mRNAs and 3 miRNAs for the second data set, associated with adjusted *P*-values less than 0.01, respectively.

We try to estimate the coincidence of genes between TD and PP; Tables 1–4 list the comparisons of genes between TD-based unsupervised FE and PP, Eqs (30) or (31) and demonstrate a high coincidence with each other. Fig 4 show the comparisons of $P_i$ and $P_k$ between TD-based unsupervised FE and PP, Eqs (30) or (31). It is obvious that smaller *P*-values used for gene selection as well as the overall distributions of *P*-values are coincident between TD-based unsupervised FE and PP, Eqs (30) or (31).

**Equivalence between K-means and PCA.** To understand these excellent and unexpected coincidences between TD-based unsupervised FE and PP, we first considered the relationship between PCA and PP and later related it with TD. PCA was known to be equivalent to K-means [3]; the space spanned by centroids of optimal sample clusters can be reproduced by the PC score attributed to the features. Suppose that we have $x_{ij} \in \mathbb{R}^{N \times M}$ which is the value of the *i*th feature of the *j*th sample. *M* samples are supposed to be clustered into *S* clusters. The centroid of *s*th cluster, $\boldsymbol{m}_s \in \mathbb{R}^N$ is defined as

$$\boldsymbol{m}_s = \frac{1}{n_s}\sum_{j \in C_s} \boldsymbol{x}_j \tag{32}$$

**Table 3. Confusion matrix of selected mRNAs between TD-based unsupervised FE and PP in the second data set.** *P*-value computed by Fisher's exact test is 0.0 within numerical accuracy (i.e., smaller than the possible smallest number given numerical accuracy).

| | | PP | |
|---|---|---|---|
| | | adjusted $P_i > 0.01$ | adjusted $P_i < 0.01$ |
| TD based unsupervised FE | adjusted $P_i > 0.01$ | 33781 | 8 |
| | adjusted $P_i < 0.01$ | 23 | 186 |

**Table 4. Confusion matrix of selected miRNAs between TD based unsupervised FE and PP in the second data set.** *P*-value computed by Fisher's exact test is $1.87 \times 10^{-7}$.

| | | PP | |
|---|---|---|---|
| | | adjusted $P_k > 0.01$ | adjusted $P_k < 0.01$ |
| TD based unsupervised FE | adjusted $P_k > 0.01$ | 316 | 0 |
| | adjusted $P_k < 0.01$ | 0 | 3 |

where $\boldsymbol{x}_j = (x_{1j}, x_{2j}, \cdots, x_{Nj})^T \in \mathbb{R}^N$, $C_s$ is a set of *j*s that belong to the *s*th cluster, $n_s$ is the size of the *s*th cluster. Here we define the projection of any vector $\boldsymbol{x} \in \mathbb{R}^N$ onto the centroid subspace as

$$S_b \boldsymbol{x} = \sum_{s=1}^{S} n_s (\boldsymbol{m}_s^T \cdot \boldsymbol{x}) \qquad (33)$$

where

$$S_b = \sum_{s=1}^{S} n_s \boldsymbol{m}_s \otimes \boldsymbol{m}_2^T \in \mathbb{R}^{N \times N} \qquad (34)$$

where $\otimes$ is the Kronecker product. $S_b$ is also known to be represented as

$$S_b = \sum_{s=1}^{S} X\boldsymbol{h}_s \otimes \boldsymbol{h}_s^T X^T = X \left( \sum_{s=1}^{S} \boldsymbol{h}_s \otimes \boldsymbol{h}_s^T \right) X^T \qquad (35)$$

where $\boldsymbol{h}_s \in \mathbb{R}^M$ is

$$\boldsymbol{h}_{js} = \begin{cases} \dfrac{1}{\sqrt{n_s}} & j \in C_s \\ 0 & j \notin C_s \end{cases}, \qquad (36)$$

which take non-zero values only when the *j*th sample belongs to the *s*th cluster. K-means is an algorithm to find clusters that minimize

$$J_S = \sum_{s=1}^{S} \sum_{j \in C_s} (\boldsymbol{x}_j - \boldsymbol{m}_s)^2. \qquad (37)$$

Minimization of $J_k$ is known to be equivalent to the maximization of $\text{Tr}S_b$, which means the trace of matrix $S_b$. It is known that

$$\min_{\{\boldsymbol{h}_s\}} S_b = \sum_{\ell=1}^{S-1} \lambda_\ell \boldsymbol{u}_\ell \otimes \boldsymbol{u}_\ell^T \qquad (38)$$

where $\boldsymbol{u}_\ell \in \mathbb{R}^N$ is the vector whose components are $\ell$th PC scores attributed to the features and eigenvector of the gram matrix as

$$XX^T \boldsymbol{u}_\ell = \lambda_\ell \boldsymbol{u}_\ell. \qquad (39)$$

If we compare Eq (35) with Eq (38), we can notice that $\sum_{s=1}^{S} X\boldsymbol{h}_s \otimes \boldsymbol{h}_s^T X^T$ corresponds to $\sum_{\ell=1}^{S-1} \lambda_\ell \boldsymbol{u}_\ell \otimes \boldsymbol{u}_\ell^T$, and PCA can give us an optimal centroid subspace, $S_b$, even without realizing the clusters by K-means, i.e., in a fully unsupervised manner.

At first, when the clusters are the solution of K-means, the centroid subspace can be represented by the PC score which can also be expressed by $X \boldsymbol{h}_s$. $\boldsymbol{h}_s$ is clearly coincident with $y_j$ defined in Eq (27). This means that PP employing $\boldsymbol{u}_\ell$ as $\boldsymbol{b}$ should result in projection onto the centroid subspace when $y_j$ is coincident with the clusters. Here we define $y_j$, Eq (27), such that it can represent distinction between tumors and normal tissues, which should be detected by K-means. This explains why TD-based unsupervised FE works well and why PP can be replaced with TD-based unsupervised FE. To our knowledge, this is the first rationalization on why TD- and PCA-based unsupervised FE work well.

One might wonder whether the above explanation is applicable to PCA while TD was applied to the first and second data sets. This gap can be explained as follows. Tensor $x_{ijk}$, was generated as the product of $x_{ij}$ and $x_{kj}$. Suppose these two are decomposed as

$$x_{ij} = \sum_\ell \lambda_\ell u_{\ell i} v_{\ell j} \tag{40}$$

$$x_{kj} = \sum_\ell \lambda'_\ell u_{\ell k} v'_{\ell j}. \tag{41}$$

If $v_{\ell j} = v'_{\ell j}$ then

$$x_{ik} = \sum_j x_{ij} x_{jk} = \sum_j \sum_\ell \lambda_\ell u_{\ell i} v_{\ell j} \sum_{\ell'} \lambda'_{\ell'} u_{\ell' k} v_{\ell' j} \tag{42}$$

$$= \sum_\ell \sum_{\ell'} \lambda'_{\ell'} \lambda_\ell u_{\ell i} u_{\ell' k} \sum_j v_{\ell j} v_{\ell' j} \tag{43}$$

$$= \sum_\ell \sum_{\ell'} \lambda'_{\ell'} \lambda_\ell u_{\ell i} \delta_{\ell \ell'} = \sum_\ell \lambda_\ell \lambda'_\ell v_{\ell i} v_{\ell k}. \tag{44}$$

This means that if $v_{\ell j} = v'_{\ell j}$, the SVD of $x_{ik}$ gives $u_{\ell i}$ and $u_{\ell k}$ that are obtained when SVD is applied to $x_{ij}$ and $x_{kj}$. Here $u_{2j}^{\mathrm{mRNA}}$ is highly correlated with $u_{2j}^{\mathrm{miRNA}}$ [12]. This is coincident with the requirement $v_{\ell j} = v'_{\ell j}$. As SVD is equivalent to PCA, this might explain why TD-based unsupervised FE works well even though the above rationalization is applied only to PCA.

**The third data set.** Next, we would like to extend the above discussion to TD. Therefore, we consider a third data set analyzed in another study [13] where we performed *in silico* drug discovery for SARS-CoV-2 by applying TD-based unsupervised FE to the gene expression profiles of human cell lines infected with SARS-CoV-2. The third data set comprises five cell lines infected with either mock (control) or SARS-Cov-2, including three biological replicates. It is formatted as tensor, $x_{ijkm} \in \mathbb{R}^{N \times 5 \times 2 \times 3}$, that represents the expression of the $i$th gene of the $j$th cell line from the infected (k = 1) or control (k = 2) group in the $m$th biological replicate. HOSVD was applied to $x_{ijkm}$ and we got

$$x_{ijkm} = \sum_{\ell_1=1}^{5} \sum_{\ell_2=1}^{2} \sum_{\ell_3=1}^{3} \sum_{\ell_4=1}^{N} G(\ell_1 \ell_2 \ell_3 \ell_4) u_{\ell_1 j} u_{\ell_2 k} u_{\ell_3 m} u_{\ell_4 i}. \tag{45}$$

In this study, we selected $\ell_1 = 1, \ell_2 = 2, \ell_3 = 1$ based on biological discussions. We then realized that $G(5, 1, 2, 1)$ has the largest absolute value given $\ell_1 = 1, \ell_2 = 2, \ell_3 = 1$. Thus, $u_{5i}$ was used to attribute $P$-values to gene $i$ using

$$P_i = P_{\chi^2} \left[ > \left( \frac{u_{5i}}{\sigma_5} \right)^2 \right] \tag{46}$$

and the obtained $P$-values were corrected using the by BH criterion; further, 163 genes associated with adjusted $P$-values less than 0.01 were selected. We now relate TD to the above discussion about PCA. Because of the HOSVD algorithm, $u_{\ell_4 i}$ can also be obtained by applying SVD to the unfolded matrix, $X \in \mathbb{R}^{N \times 30}$. Here 30 columns correspond to one of 30 combinations of $j, k, m$. Here we select $\ell_1 = 1, \ell_2 = 2, \ell_3 = 1$ so that the gene expression is independent of the cell lines and biological replicates and has opposite signs between the control and infected cells. Thus, two clusters are expected, each of which corresponds to either the control or infected cell lines. The reason why $\ell_4 = 5$ is selected is simply because $u_{5i}$ is composed of the centroid subspace coincident with two clusters. Thus, in this sense, the above discussion about PCA can be directly applied to this result.

To confirm this, $y_j$ was taken to be

$$y_{jkm} = \alpha_j \beta_k \gamma_m \tag{47}$$

$$\alpha_j = 1 \tag{48}$$

$$\beta_k = (-1)^k \tag{49}$$

$$\gamma_m = 1 \tag{50}$$

such that it represented the distinction between $k = 1$ and $k = 2$ (i.e. that between infected and control cell lines), where $y_{jkm} \in \mathbb{N}^{5 \times 2 \times 3}, \alpha_j \in \mathbb{N}^5, \beta_k \in \mathbb{N}^2$, and $\gamma_m \in \mathbb{N}^3$. Then PP was performed as

$$b_i = \sum_{j,k,m} x_{ijkm} y_{jkm}. \tag{51}$$

$P$-values were attributed to genes as

$$P_i = P_{\chi^2}\left[> \left(\frac{b_i}{\sigma_b}\right)^2\right] \tag{52}$$

and 155 genes associated with corrected $P$-values less than 0.01 were selected, where $b_i$ is expected to play a role of $u_{5i}$ in Eq (46). Table 5 lists high coincidence of selected genes between TD-based unsupervised FE and PP. Fig 5 shows the overall coincidence of distributions of $P$-values between TD-based unsupervised FE and PP. Thus, why TD based unsupervised FE can work well is explained by the ability of singular value vectors to generate a centroid subspace of clusters coincident with control and infected cell lines.

One might wonder why TD is needed if $u_{\ell_4 i}$ can be computed by applying SVD to the unfolded matrix. To understand this, we compared $v_{5(ijk)}$ obtained by applying SVD to an unfolded matrix, and corresponding to $u_{5i}$ as well as $u_{1j} u_{2k} u_{1m}$ with $y_{ikm}$. While $u_{1j} u_{2k} u_{1m}$ is well coincident with $y_{jkm}$, $v_{5(jkm)}$ is not (Fig 8). Thus, we need to apply TD to $x_{ijkm}$ to obtain singular value vectors attributed to samples, which are coincident with two clusters but cannot be obtained when SVD is applied to an unfolded matrix.

**Table 5. Confusion matrix of selected genes between TD-based unsupervised FE and PP in the third data set.** $P$-value computed by Fisher's exact test is $1.40 \times 10^{-241}$.

| | | PP | |
|---|---|---|---|
| | | adjusted $P_i > 0.01$ | adjusted $P_i < 0.01$ |
| TD based unsupervised FE | adjusted $P_i > 0.01$ | 21582 | 52 |
| | adjusted $P_i < 0.01$ | 60 | 103 |

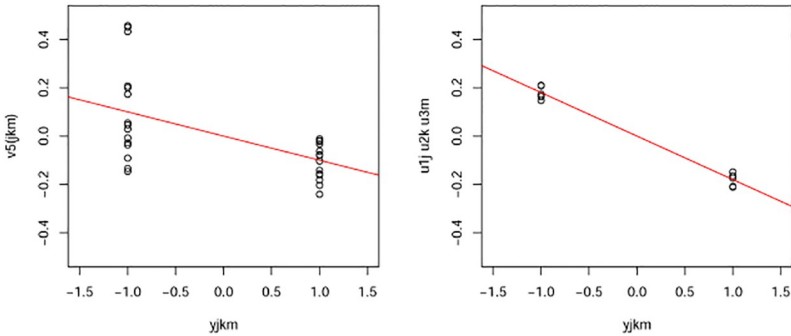

**Fig 8. Comparisons between $y_{jkm}$ and either $v_{5(jkm)}$ or $u_{1j}u_{2k}u_{1m}$.** Red straight lines indicate linear regressions.

**Rationalization of threshold *P*-values.** As we have successfully shown that TD as well as PCA are equivalent to PP that aims to maximize projection onto the subspace centroid of clusters coincident with the desired distinction (cancer vs. normal tissue or control vs. infected cell lines), we would next like to rationalize the *P*-values computed by the $\chi^2$ distribution and threshold values of $P = 0.01$, which have long been employed to select DEGs with PCA- and TD-based unsupervised FE. Because distribution of projection in the infinite sample number limits is proven to be always Gaussian [6], this null hypothesis might seem reasonable. Nonetheless, the individual distribution of gene expression is far from Gaussian and is rather close to negative signed binomial distribution and when the number of samples is not large enough, the distribution of projection does not converge with a Gaussian distribution at all. Thus, a more straightforward rationalization is needed. Therefore, we generated a null distribution by shuffling *i* in each sample and recomputed the singular value vectors, $u_{\ell_1 i}$ (for mRNA in the first and the second data sets), $u_{\ell_3 k}$ (for miRNA in the first and the second data sets), and $u_{\ell_5 i}$ (for genes in the third data set). Then *P*-values were recomputed using the generated null distribution and were corrected using the BH criterion to obtain genes associated with significant adjusted *P*-values. In the following, we apply the shuffling to three data sets, the first, the second, and the third data set, and select genes using *P*-values obtained by shuffling. Coincidence of selected genes and distribution of *P*-values between PCA or TD and shuffling is estimated. These evaluations enable us to discuss the suitability of threshold *P*-values.

Fig 1(A) shows the histogram of raw *P*-values computed using the null distribution generated by shuffling one hundred times when the miRNAs in the first data set were considered. As it is obvious that there are too many *P*-values near 1, we excluded some miRNAs with low values to obtain a *P*-value distribution more coincident with the null distribution. Fig 1(B) shows the histogram of raw *P*-values computed to be restricted to the top 500 more expressive miRNAs; this seems more coincident with the null distribution. We then found that twelve miRNAs are associated with adjusted *P*-values less than 0.1. Table 6 lists the comparison of selected miRNAs between TD-based unsupervised FE and the null distribution generated by shuffling. Although the threshold *P*-values differ between the two, the selected miRNAs are quite coincident. A threshold *P*-value 0.01 was empirically employed for PCA- and TD-based unsupervised FE as it often gave us biologically reasonable results. $P = 0.01$ in Gaussian distribution is assumed as the null hypothesis corresponding to $P = 0.1$ when the null distribution is generated by shuffling. Although this discrepancy must be fulfilled in the future, we conclude that their performances are quite similar.

Fig 2(A) shows the histogram of raw *P*-values computed using the null distribution generated by shuffling one hundred times when mRNAs in the first data set were considered. As it is

**Table 6. Confusion matrix of selected miRNAs between TD-based unsupervised FE and shuffling in the first data set.** *P*-value computed by Fisher's exact test is $1.28 \times 10^{-21}$.

| | | shuffling | |
|---|---|---|---|
| | | adjusted $P_k > 0.1$ | adjusted $P_k < 0.1$ |
| TD based unsupervised FE | adjusted $P_k > 0.01$ | 488 | 1 |
| | adjusted $P_k < 0.01$ | 0 | 11 |

**Table 7. Confusion matrix of selected mRNAs between TD-based unsupervised FE and shuffling in the first data set.** *P*-value computed by Fisher's exact test is $2.69 \times 10^{-137}$.

| | | shuffling | |
|---|---|---|---|
| | | adjusted $P_i > 0.1$ | adjusted $P_i < 0.1$ |
| TD based unsupervised FE | adjusted $P_i > 0.01$ | 2928 | 0 |
| | adjusted $P_i < 0.01$ | 3 | 69 |

obvious that there are too many *P*-values near 1, we excluded some mRNAs with low values to obtain a *P*-value distribution more coincident with the null distribution. Fig 2(B) shows the histogram of raw *P*-values computed to be restricted to the top 3000 more expressive mRNAs; this seems more coincident with the null distribution. We then found that 69 mRNAs are associated with adjusted *P*-values less than 0.1. Table 7 lists the comparison of selected mRNAs between TD-based unsupervised FE and the null distribution generated by shuffling. Although threshold *P*-values differ between the two, the selected mRNAs are quite coincident. A threshold *P*-value 0.01 was empirically employed for PCA- and TD-based unsupervised FE as it often gave us biologically reasonable results. *P* = 0.01 in a Gaussian distribution is assumed as the null hypothesis corresponding to *P* = 0.1 when the null distribution is generated by shuffling. Although this discrepancy must be fulfilled in the future, we conclude that their performances are quite similar.

Fig 6(A) shows the histogram of raw *P*-values computed using the null distribution generated by shuffling one hundred times when miRNAs in the second data set were considered. As it is unlikely to get significant *P*-values, we did not select miRNAs associated with significant *P*-values. Fig 6(B) shows the histogram of raw *P*-values computed for mRNAs in the second data set; there are no peaks around *P* = 1. We then found that 262 mRNAs are associated with adjusted *P*-values less than 0.1. Table 8 lists the comparison of selected mRNAs between TD-based unsupervised FE and the null distribution generated by shuffling. Although threshold *P*-values differ between the two, the selected mRNAs are well coincident. A threshold *P*-value 0.01 was empirically employed for PCA- and TD-based unsupervised FE as it often gave us biologically reasonable results. *P* = 0.01 in a Gaussian distribution is assumed as the null hypothesis corresponding to *P* = 0.1 when the null distribution is generated by shuffling. Although this discrepancy must be fulfilled in the future, we conclude that their performances are quite similar.

**Table 8. Confusion matrix of selected mRNAs between TD-based unsupervised FE and shuffling in the second data set.** *P*-value computed by Fisher's exact test is 0.0 within numerical accuracy (i.e., smaller than the possible smallest number given numerical accuracy).

| | | shuffling | |
|---|---|---|---|
| | | adjusted $P_i > 0.1$ | adjusted $P_i < 0.1$ |
| TD based unsupervised FE | adjusted $P_i > 0.01$ | 33736 | 53 |
| | adjusted $P_i < 0.01$ | 0 | 209 |

**Table 9. Confusion matrix of selected genes between TD-based unsupervised FE and shuffling in the third data set.** *P*-value computed by Fisher's exact test is $5.00 \times 10^{-63}$.

| | | shuffling | |
|---|---|---|---|
| | | adjusted $P_i > 0.1$ | adjusted $P_i < 0.1$ |
| TD based unsupervised FE | adjusted $P_i > 0.01$ | 2617 | 0 |
| | adjusted $P_i < 0.01$ | 115 | 48 |

Fig 3(A) shows the histogram of raw *P*-values computed using the null distribution generated by shuffling one hundred times when considering the genes in the third data set. As there were too many *P*-values less than 0.2, we excluded some mRNAs with low values to obtain a *P*-value distribution more coincident with the null distribution. Fig 3(B) shows the histogram of raw *P*-values computed to be restricted to the top 2780 more expressive mRNAs; this seems more coincident with the null distribution. We then found that 48 mRNAs are associated with adjusted *P*-values less than 0.1. Table 9 lists the comparison of selected mRNAs between TD-based unsupervised FE and the null distribution generated by shuffling. Although threshold *P*-values differ between two, selected mRNAs are well coincident. A threshold *P*-value 0.01 was empirically employed for PCA- and TD-based unsupervised FE as it often gave us biologically reasonable results. *P* = 0.01 in a Gaussian distribution is assumed as the null hypothesis corresponding to *P* = 0.1 when the null distribution is generated by shuffling. Although this discrepancy must be fulfilled in the future, we conclude that their performances are quite similar.

## Discussion

In the previous section, we explained why PCA- and TD-based unsupervised FE work well (because singular value vectors correspond to projection onto the centroid subspace obtained by K-means) and how the criterion to select genes associated with adjusted *P*-values less than 0.01, which was computed assuming the null hypothesis that singular value vectors obey Gaussian distribution, is empirically coincident with another criterion to select the genes associated with adjusted *P*-values less than 0.1, which are computed assuming the null distribution generated by shuffling.

There are many points to be discussed. In the above example, we only dealt with the case wherein only two clusters could be distinguished in a one-dimensional space (i.e., only one singular value vector). Considering cases with more clusters might be challenging, projections onto subspace centroids do not have a one-to-one correspondence with singular value vectors as the coincidence between the projection to the subspace centroid and singular value vectors stands only between the spaces spanned by them, and not between themselves. Despite this, TD- and PCA-based unsupervised FE applied to more than two classes is known to work rather as well as in the case with only two clusters [16].

On the contrary, although we could only discuss cases with a finite number of clusters, PCA- and TD-based unsupervised FE are also known to work in detecting parameter dependence, e.g., time development [17, 18]. Extending the discussion here to regression analysis without any clusters will be the next step.

One might also wonder whether we need TD if singular value vectors attributed to genes are common between TD and PCA. At first, in the integrated analysis of mRNA and miRNA, TD-based unsupervised FE could outperform PCA-based unsupervised FE [12]. Similarly, TD-based unsupervised FE outperformed PCA-based unsupervised FE in the integrated analysis of gene expression and DNA methylation [19]. Thus, TD-based unsupervised FE is required when integrated analysis is targeted. Even when no integrated analysis was targeted,

TD based unsupervised FE can give singular value vectors that are more coincident with biological clusters (Fig 8). Thus, despite the apparent equality of singular value vectors attributed to genes between TD and PCA, TD-based unsupervised FE is a more useful strategy than PCA-based unsupervised FE.

Although we did not clearly denote this, conventional gene selection strategies based on statistical tests are known to fail when applied to the first, second, and third data sets [12, 13]; they always selected too many or too few genes, mRNAs, and miRNA, which is in contrast to TD-based unsupervised FE that could always select a restricted number of genes, from tens to hundreds.

One might also wonder why we did not employ the null distribution generated by shuffling instead of the un-justified Gaussian distribution, with PCA- and TD-based unsupervised FE. As can be seen above, employment of null distribution generated by shuffling is not straightforward; in some cases, e.g, the first and the third data sets mentioned above, we needed to exclude low expressed genes manually whereas this was not required for the second data set. No miRNAs that were significantly expressed distinctly between controls and cancers in the second data sets were detected with the null distribution generated by shuffling. In addition, the number of low expressed genes to be removed cannot be decided uniquely. On the contrary, the criterion that genes associated with adjusted $P$-values less than 0.01 assuming the null hypothesis that singular value vectors obey a Gaussian distribution is more robust. This often can give a restricted number of genes without excluding low expressed genes. Although why this works so well must be explored in the future, it is an empirically more useful strategy than the null distributions generated by shuffling.

One may also wonder why we did not employ the centroid subspace, $S_b$, instead of singular value vectors if these two are equivalent for optimal clusters and the meaning of centroid subspace is easier to understand compared to singular value vectors. At first, we needed to apply K-means which often fail in unbalanced data sets composed of clusters with a very distinct number of samples. Next, K-means always identifies the primary cluster. Nevertheless, in the case of SARS-CoV-2 (the third data set), distinction between infected cell lines and control cell lines was detected using the fifth singular value vectors whose contribution will probably be neglected by K-means because of its too small contribution. In addition, singular value vectors can be computed in a fully unsupervised manner that does not require any labeling. Considering these advantages, it is reasonable to use singular value vectors instead of a centroid subspace despite its apparent usefulness. Further, as the $y_j$ used to compute projection $\boldsymbol{b}$ is decided manually, even if some biological features that $y_j$ assumes, such as clusters, do not exist, $\boldsymbol{b}$ can be computed. This might result in wrong conclusions. However, if there are no clusters at all, because no corresponding singular value vectors attributed to samples and coincident with $y_j$ are obtained, we can have an opportunity to realize any misunderstanding. Thus, usage of singular value vectors but not projection $\boldsymbol{b}$ might be advantageous.

One might also wonder why other more frequently used TD such as CP decomposition [3] have not been employed instead of HOSVD. This might be understood as follows. In the above description, we could relate the singular value vectors obtained by HOSVD to the centroid subspace, because singular value vectors attributed to genes are common between HOSVD and PCA. This equivalence will be broken if HOSVD is replaced with other TDs. When we invented TD-based unsupervised FE, though we also tested other TDs [3], HOSVD always outperformed other TDs when used for feature selections. The equivalence of HOSVD and PCA might explain why HOSVD could outperform other popular TDs as a feature selection tool.

Another possible concern is that only one hundred times shuffling was performed for the computation in Figs 1 to 3 whereas we considered $P$-values equal to 0.01; nevertheless, it is not

problematic at all because of the following two reasons. First of all, the *P*-values we considered were not raw *P*-values but corrected *P*-values. Thus total number of probabilities computed are much larger than one hundred. Since the numbers of computed *P*-values are as many as those of mRNAs and miRNAs, they are as many as $10^3$ or $10^4$. Thus, the number of shuffling, one hundred, is not directly related to *P*-values of 0.01 at all. Second, individual *P*-values are not related to the number of shuffling at all; what we have performed was to generate *P*-values whose number is equal to that of miRNAs or mRNAs, i.e., $10^3$ or $10^4$. Thus, individual *P*-values can take much smaller values than 0.01, say $10^{-3}$ and $10^{-4}$ for miRNAs and mRNAs, respectively. Increasing or decreasing the number of shuffling does not affect the absolute values of *P*-values at all. The number of shuffling is only related to the reproducibility; if we can compute *P*-values based upon only one shuffling, it might heavily fluctuate. On the other hand, if we take average of *P*-values over one hundred shuffling, their outcome is expected to be more stable. The purpose of taking average over one hundred shuffling is simply because of stability of outcome. Apparent relationship between *P* = 0.01 and one hundred times shuffling does not make any sense. In conclusion, even if we take *P* = 0.01 as a threshold for one hundred times shuffling, it is not a problem at all.

Based upon the studies presented in the above, we emphasize that the usages of PCA or TD based unsupervised FE are recommended, since generally we do not know to which direction we project the data sets. PCA and TD turned out to have ability to give the directions of projections in an unsupervised manner. When projections directions are trivial, e.g., distinction between two classes, PCA and TD can correctly give us the directions. Even if the data sets are more complicated, we can employ higher mode tensors to tackle more complicated data sets. PCA and TD based unsupervised methods will be promising methods.

## Author Contributions

**Conceptualization:** Y-h. Taguchi.

**Data curation:** Turki Turki.

**Formal analysis:** Y-h. Taguchi.

**Methodology:** Y-h. Taguchi.

**Software:** Y-h. Taguchi.

**Supervision:** Y-h. Taguchi.

**Writing – original draft:** Y-h. Taguchi, Turki Turki.

**Writing – review & editing:** Y-h. Taguchi, Turki Turki.

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
