## [Decision Letter · Decision Letter 0]

23 Aug 2022

PONE-D-22-20332Projection in genomic analysis: A theoretical basis to rationalize tensor decomposition and principal component analysis as feature selection toolsPLOS ONE

Dear Dr. Taguchi,

Thank you for submitting your manuscript to PLOS ONE. After careful consideration, we feel that it has merit but does not fully meet PLOS ONE’s publication criteria as it currently stands. Therefore, we invite you to submit a revised version of the manuscript that addresses the points raised during the review process.

We look forward to receiving your revised manuscript.

Kind regards,

Chi-Hua Chen, Ph.D.

Academic Editor

PLOS ONE

Journal Requirements:

"This work was supported by KAKENHI [grant numbers 19H05270, 20H04848, and 20K12067] to YT and Institutional Fund Project (IFPIP) from the Ministry of Education and King Abdulaziz University (DSR), Jeddah, Saudi Arabia [grant number IFPIP: 924-611-1442] to TT."

Reviewers' comments:

Reviewer's Responses to Questions

**Comments to the Author**

1. Is the manuscript technically sound, and do the data support the conclusions?

Reviewer #1: Yes

Reviewer #2: Yes

2. Has the statistical analysis been performed appropriately and rigorously? 

Reviewer #1: Yes

Reviewer #2: Yes

3. Have the authors made all data underlying the findings in their manuscript fully available?

Reviewer #1: Yes

Reviewer #2: No

4. Is the manuscript presented in an intelligible fashion and written in standard English?

Reviewer #1: Yes

Reviewer #2: Yes

5. Review Comments to the Author

Reviewer #1: The paper analyzes the reason why the recently proposed principal component analysis (PCA) and tensor decomposition (TD)-based unsupervised feature extraction (FE) has often outperformed these statistical test-based methods in the context of projection pursuit that was proposed a long time ago. Some findings in this paper rationalize the success of PCA- and TD-based unsupervised FE for the first time. I have the following suggestions for this manuscript. Other comments can see the attached file.

Reviewer #2: General Comments:

Is the paper new, technically correct, and relevant?

Yes, the paper is new and technically sounds. Results somehow does support the methodology, but needed to be more cleared by the author in case of properties of the data.

Is the paper well organized?

The paper is properly organized, good literature review, suitable motivation and clear explanation on results are positive points to that.

Is the abstract concise?

Yes, but I think it needs to be rephrased after revision to add some comments about any artifacts or negative points in the method, if exist.

Is the introduction motivating?

Yes, Introduction section is motivating.

Are the methodology, results, and conclusions completely developed?

No, they need to be modified and developed according to the technical comments.

Are there language, mathematics, reference, or style errors? There is no mathematical, reference or style error.

Technical Comments:

Are the codes available for this research? As I found, there is no code available for this study, e. g. in Github. If the authors could make the codes available, the manuscript could be much better evaluated, not only for reviewers, but also for possible readers. When it is not possible to upload the code for public access, such as in Github, could they be provided for reviewer for better assessment of the study?

The study is comprehensive and requires large time to be read carefully and being reviewed. The theoretical background has been well explained in details, and the experiments and related models are presented and the algorithm in Fig. 1 is also well presented. I think more explanation about the steps and the parameters in Fig. 1 is required.

The result comparison parts are well organized and presented. The display way is good. But quantitative evaluation is somehow too much that one can get lost in that. I think it would be better that you add more explanation to that.

How did you evaluate the final result? How did you consider to finally selection a methodology for the most complicate problem?

What about when the models are more complex?

The introduction section is a nice one. It is architected very beautifully, while written fully academic and comprehend. I assume that any change in the introduction section is not necessary, but one of the important tasks after publishing a study is to increase its chance to be seen by the most possible number of researchers, so I would like to give two recommendations. First, to get your published study in the list of searched for papers based on keywords, I propose to increase variety of your keywords. In my viewpoint, they do not cover the whole topic of the study and are not widely searched words. I propose to add at least the keyword “data analysis”. Second, one of the methods in the publisher’s website that brings a publication on to the researchers, is based on the similar publications that they have read before. So, the more you cite similar publication, the more the chance that the search engine in the publisher website propose your paper to the researcher. Besides of that, it will also complete your introduction section. As another advantage, it rises new ideas to the researchers by combining various methods, or resolving drawback of one seen paper by reading the similar one, or extending the methodology to a fully automatic one. So, based on these points, I would like to ask to cite to the following similar publication in the manuscript which used PCA and feature selection for deep learning, but in different field of study. The first proposed publication is: Shahbazi, A., Soleimani Monfared, M., Thiruchelvam, V., Ka Fei, T., Babasafari, A.A., (2020). Integration of knowledge-based seismic inversion and sedimentological investigations for heterogeneous reservoir. Journal of Asian Earth Sciences. The second publication for citation is: Khayer, K., Kahoo, A.R., Soleimani Monfared, M., Tokhmechi, B., and Kavousi, K., (2022). Target-Oriented Fusion of Attributes in Data Level for Salt Dome Geobody Delineation in Seismic Data. Natural resource research, and the other publication could be: Khayer, K., Kahoo, A.R., Soleimani Monfared, M., and Kavouosi, K., (2022). Combination of seismic attributes using graph-based methods to identify the salt dome boundary. Journal of Petroleum Science and Engineering. 215, Part A, 110625,

The abstract focusses mainly on the general problem and ignores the other items of the abstract such as the methodology, good introduction, results and conclusion.

The authors should explain what limitations did they find out about the proposed method.

Best Regard

6. PLOS authors have the option to publish the peer review history of their article (what does this mean?). If published, this will include your full peer review and any attached files.

Reviewer #1: No

Reviewer #2: No

---

## [Decision Letter · Decision Letter 1]

19 Sep 2022

Projection in genomic analysis: A theoretical basis to rationalize tensor decomposition and principal component analysis as feature selection tools

PONE-D-22-20332R1

Dear Dr. Taguchi,

We’re pleased to inform you that your manuscript has been judged scientifically suitable for publication and will be formally accepted for publication once it meets all outstanding technical requirements.

Kind regards,

Chi-Hua Chen, Ph.D.

Academic Editor

PLOS ONE

Additional Editor Comments (optional):

Reviewers' comments:

Reviewer's Responses to Questions

**Comments to the Author**

1. If the authors have adequately addressed your comments raised in a previous round of review and you feel that this manuscript is now acceptable for publication, you may indicate that here to bypass the “Comments to the Author” section, enter your conflict of interest statement in the “Confidential to Editor” section, and submit your "Accept" recommendation.

Reviewer #1: All comments have been addressed

Reviewer #2: All comments have been addressed

2. Is the manuscript technically sound, and do the data support the conclusions?

Reviewer #1: Yes

Reviewer #2: Yes

3. Has the statistical analysis been performed appropriately and rigorously? 

Reviewer #1: Yes

Reviewer #2: Yes

4. Have the authors made all data underlying the findings in their manuscript fully available?

Reviewer #1: Yes

Reviewer #2: Yes

5. Is the manuscript presented in an intelligible fashion and written in standard English?

Reviewer #1: Yes

Reviewer #2: Yes

6. Review Comments to the Author

Reviewer #1: This manuscript has enriched the content of the article and enhanced the readability of the article through modification, but there are still some small problems.

1. It is suggested that the paragraphs of the full paper should be aligned at both ends, which may make the article look more beautiful.

2. In line 201 on page 8, u3k does not exist in (26).

3. In line 284 on page 11, a sentence uses two verbs, “P-values were attributed to genes as... 155 genes associated with corrected P-values less than 0.01 were selected, bi is expected to play a role of u5i in eq. (46).”

4. Please check the references carefully. For example, reference [3], [10], [12], [14], [15], [16], and [19] etc.

Reviewer #2: Dear Authors;

I have read your response and edited manuscript carefully and I was pleased with your answers and the way of developing the research and the manuscript.

So, I have no further comment for you.

Best Regards

7. PLOS authors have the option to publish the peer review history of their article (what does this mean?). If published, this will include your full peer review and any attached files.

Reviewer #1: No

Reviewer #2: No

---

## [Editor Report · Acceptance letter]

20 Sep 2022

PONE-D-22-20332R1 

Projection in genomic analysis: A theoretical basis to rationalize tensor decomposition and principal component analysis as feature selection tools 

Dear Dr. Taguchi:

I'm pleased to inform you that your manuscript has been deemed suitable for publication in PLOS ONE. Congratulations! Your manuscript is now with our production department. 

Kind regards, 

on behalf of

Professor Chi-Hua Chen 

Academic Editor

PLOS ONE